

# Molecular characterization of clonal lineage and staphylococcal toxin genes from *S. aureus* in Southern Nigeria

Funmilola A. Ayeni[1], Werner Ruppitsch[2] and Franz Allerberger[2]

[1] Department of Pharmaceutical Microbiology, Faculty of Pharmacy, University of Ibadan, Ibadan, Nigeria
[2] Division of Human Medicine, Austrian Agency for Health and Food Safety, Vienna, Austria

Corresponding author
Funmilola A. Ayeni,
funmiyeni@yahoo.co.uk

## ABSTRACT

**Background.** *Staphylococcus aureus* is a human colonizer with high potential for virulence, and the spread of the virulent strains from the colonized hosts to non-carriers in the community is on the increase. However, there are few reports on comprehensive analysis of staphylococcal enterotoxin (SE) genes with clonal lineage in *S. aureus* in Africa. This is essential because of diversity of cultures and habits of the people. This study analyzed spa types and enterotoxin genes in *S. aureus* strains previously isolated from the human nostrils, poultry and clinical samples in Southern Nigeria.

**Methods.** Forty-seven *S. aureus* isolates were obtained from humans nostrils ($n = 13$), clinical strains ($n = 21$) and poultry ($n = 13$) from previous studies in Southern Nigeria. The strains were analyzed for *mecA* gene, selected toxins genes (*sea, seb, sec, sed, see, seg, seh, sei, sej, sek, sel, sem, sen, seo, sep, seq, ser, seu)* and Panton-Valentine leukocidin (PVL) gene *(*lukS-PV/lukF-PV*)* by PCR. Population structures of the strains were detected by Staphylococcal protein A (*spa*) typing.

**Results.** Twenty different spa types were obtained with the highest percentages, 17% observed in *spa* type t091 from clinical, nasal and poultry samples while t069 was the most prevalent spa type in poultry. Two MRSA were only detected in human strains. The poultry strains had the highest occurrence of SE genes (18%) followed by nasal strains (15%) and clinical strains (10%). Eighty-nine percent of all tested isolates harbored at least one SE gene; *seo* was the most prevalent (34%) followed by *seg* (30%) and *sea* (21%), while *sec, see* and *sej* were absent in all strains. Spa type t355 was associated with *lukS-PV/lukF-PV* gene and complete absence of all studied SE. *Sea, seq, seb, sek* were associated with spa type t069; *sea* was associated with t127 while *sep* was associated with spa type t091. There were coexistences of *seo/seg* and *sei/seg*.

**Conclusions.** The higher carriage of staphylococci enterotoxin genes by the nasal and poultry *S. aureus* strains suggests a high potential of spread of staphylococcal food poisoning through poultry and healthy carriers in the community. This is the first report of high occurrence of staphylococcal enterotoxins genes in poultry from Nigeria.

## BACKGROUND

*Staphylococcus aureus* is a major pathogen affecting healthy and hospitalized human, and also occurs in animals. It is one of the most important human colonizers implicated

in infectious diseases. Colonization of human nares by *S. aureus* is a risk factor for staphylococcal diseases (*Wertheim et al., 2004*) and invasive staphylococci infection can have its source in strains occurring naturally in the host.

The emergence of MRSA complicated the treatment of staphylococci infections and increases the focus on *S. aureus* with its broad spectrums of inherent virulence factors that enhances its capacity for infections ranging from mild skin infections to severe sepsis, pneumonia, osteomyelitis and endocarditis (*Ayepola et al., 2015*). The ability of *S. aureus* to successfully infect man is largely due to the expression of virulence factors e.g., staphylococcal enterotoxins (SE), Panton-Valentine Leucocidin (PVL) and Toxic Shock Syndrome Toxin (TSST) which promote adhesion, acquisition of nutrients and evasion of the host's immunologic responses (*Monday & Bohach, 1999*). PVL can be implicated in skin and soft tissue infections and can also increase *S. aureus*' ability to cause severe infections in humans. Staphylococcal enterotoxins are produced by *S. aureus* which enhances its status as important food-borne pathogens (*Løvseth, Loncarevic & Berdal, 2004*). The SE toxins' genes in *S. aureus* encode different virulence factors which if expressed, can produce the corresponding enterotoxins.

*S. aureus* can also colonize animals and carriage of toxigenic genes in *S. aureus* in food animals is creating concerns that pathogenic *S. aureus* strains can be transmitted through the food chain (*Vitale et al., 2015*). In pastoral Africa, there is constant contact with poultry as they are sometimes reared near the house in a free range system. Virulence genes carrying *S. aureus* may become an emerging zoonotic issue because *S. aureus* strains circulating these SE genes can be transmitted from animals to humans, thereby presenting a public health problem. The persistence of *S. aureus* is about 20% in the general population with about 60% being intermittent carriers (*Kluytmans, Van Belkum & Verbrugh, 1997*) also increase the risk of infections with higher frequency of infections in carriers than in non-carriers (*Ayepola et al., 2015*). Infections in non-carriers are commonly through contaminated food that had passed through a carrier (*Pinchuk, Beswick & Reyes, 2010*).

*Spa* typing of *S. aureus* strains provides information which can group isolates in clonal lineages. Clonal analyses can also provide useful insights into the virulence potentials and nature of *S. aureus* populations (*Kolawole et al., 2013*) which is important for the detection of transmission routes and monitoring of bacterial strains circulation. To establish better infection control in Nigeria, it is important to understand the local epidemiology and clonal lineages of *S. aureus* in the country. *Shittu et al. (2011)* reported that *S. aureus* is the main etiological agent of many infections in sub-Saharan Africa and one of the most frequently encountered bacterial species in microbiology laboratories in Nigeria while *Ayepola et al. (2015)* reported that some virulence factors were highly prevalent in *S. aureus* isolated from infection sites but less frequently found in isolates from colonization in Nigeria. Detection of the genes coding for these virulence factors will exhibit the potentials of these *S. aureus* strains being toxigenic. Therefore, the aim of this work was to detect selected staphylococci toxin genes and clonal lineage through *spa* typing of *S. aureus* previously isolated from poultry, nostrils of healthy people in the community and clinical samples in Southern Nigeria.

## METHODS

### Bacterial isolates

Forty-seven *S aureus* isolates used in this study were collected between 2011 and 2014 from our previous studies on *S. aureus* in Southwestern Nigeria. Thirteen isolates were from healthy college students living together in an hostel and 21 were from clinical samples (previously isolated from infectious wounds, urine and other body sites isolated in medical microbiology units of various hospitals) (*Ayeni, Olatunji & Ogunniran, 2014*; *Ayeni et al., 2015*; *Ayeni, Andersen & Nørskov-Lauritsen, 2017*; *Ayeni & Odumosu, 2016*) while 13 isolates were previously isolated from a poultry farm in Southwestern Nigeria. All *S. aureus* isolates from these previous studies were selected for this study.

### Identification of *S. aureus* strains by amplification of *femA* gene

The DNA of all collected staphylococci isolates was extracted by QuickExtract™ DNA extraction solution (Epicentre, Madison, WI, USA) according to the manufacturer's instructions. One µl of the extracted DNA was used in PCR reaction in a total volume of 20 µl with 10 µl of 2-fold concentrated RedTaq Ready Mix (Sigma, Darmstadt, Germany), 7 µl PCR grade water, 1 µl of 10 pmol of *femA*-F AACTGTTGGCCACTATGA and 1 µl of 10 pmol of *femA*-R CCAGCATTACCTGTAATC according to protocol previously described by *Vannuffel et al. (1995)*. The PCR product was analysed on agarose gel and bands corresponding to 686-bp were recorded as positive for *femA*. Laboratory strains were used as positive control.

### *Spa* typing of *S aureus* isolates

All *femA* positive isolates were further analysed for *spa* typing The polymorphic X region of the *spa* gene was amplified in all isolates in a total volume of 20 µl comprising 1 µl of genomic DNA, 10 µl of 2-fold concentrated RedTaq Ready Mix (Sigma, Darmstadt, Germany), 7 µl PCR grade water and 1 µl of each primer in PCR reactions according to protocol previously described (*Koreen et al., 2004*; *Schmid et al., 2013*). PCR products were analysed on 1 % agarose previously stained with GelRed (BiotiumInc, Fremont, CA, USA) at approximately 40 mAmp for 45 min. The PCR products were purified with EXOSAP-IT (GEHealthcare, UK). Two microliters of the purified PCR products were used for subsequent sequencing using the BigDye 3.1 terminator sequencing kit (Applied Biosystems, Foster City, CA, USA) and were finally analyzed on a ABI Genetic Analyzer 3500Dx (Applied Biosystems, Foster City, CA, USA). The chromatograms obtained were analyzed with Ridom Staph Type software version 1.4 (RidomGmbH, Sedanstr, Germany; http://spa.ridom.de/index.shtml). *Spa* types were deduced by the differences in number and sequence of spa repeats with the BURP algorithm (Ridom GmbH, Sedanstr, Germany) and the Ridom Spa Server database. BURP cluster analysis was performed using Ridom SpaTyper software using the parameter settings: clustering cost less or equal 5 and excluding spa types with less than five repeat units (*Koreen et al., 2004*; *Montanaro et al., 2016*).

### PCR amplification of *mecA/mecC*

PCR assay was performed for all confirmed *S. aureus* strains to amplify a region of *mecA* gene according to protocol previously described by *García-Álvarez et al. (2011)*. PCR

products were resolved by agarose (1%) gel electrophoresis previously stained with GelRed (BiotiumInc, Fremont, CA, USA) and run at approximately 40 mAmp for 45 min. The bands corresponding to expected amplicon size were recorded as positive. Laboratory strain was used as positive control.

## Virulence genes detection by PCR

PCR reactions for amplification of SE genes (*sea, seb, sec, sed, see, seg, seh, sei, sej, sek, sel, sem, sen, seo, sep, seq, ser, seu*) were performed in single PCR reactions for each gene using the following conditions: initial denaturation for 5 min at 95 °C followed by 35 cycles of denaturation (95 °C for 30 s), annealing, extension (72 °C for 1 min), and a final extension step (72 °C for 10 min).. The primer sequences, annealing temperatures and expected amplicon sizes are as described by *Monday & Bohach (1999)*, *Jarraud et al. (1999)*; *Jarraud et al. (2002)*, *Orwin et al. (2001)*, *Løvseth, Loncarevic & Berdal (2004)*. All isolates with positive bands that corresponds to the expected amplicon sizes were taken as positive for the SE gene. The assay for the lukS-PV/lukF-PV encoding the Panton-Valentine leucocidin were performed using the primers and conditions described by *Lina et al. (2003)*. Laboratory strains that were previously positive for the gene was used as positive control.

## RESULTS

Twenty different spa types were obtained from 47 confirmed *S. aureus* strains used in this study. The following spa types were detected from clinical strains: t355, t537, t1931, t1045, t021, t069, t1095, t091, t127, t008. The following was detected from nasal strains: t084, t091, t1045, t127, t939, t311, t786, t1154, while the following were detected from poultry: t069, t095, t091, t292, t939, t318, t050, t1171. Most frequent *spa* types were t091 (17%) and t355 (17%). Spa type t091 was observed from the three sources of isolation (clinical, nasal and poultry samples) while t069 is the most prevalent type in poultry (Table 1, Fig. 1). BURP clusters of the obtained spa types (Table 1) resulted in five cluster containing at least two spa types and nine singletons (Fig. 2). Of the tested strains, two were MRSA. The first MRSA strain, FAA014 is a clinical strain with spa type t069 while the second MRSA, FAA044 is a nasal strain with spa type t786 from a healthy carrier in the community (Table 2). Only one each of our nasal and poultry isolates had lukS-PV/lukF-PV while the remaining 12 isolates lacked the gene. Nine of the 21 clinical strains harbored lukS-PV/lukF-PV.

The poultry strains had the highest occurrence of SE genes (18%) followed by nasal strains (15%) and then clinical strains (10%). Eighty-nine percent of all isolates harbored at least one SE gene while *sen, sei* and *seh* were only found in nasal isolates. *Seo* was the most prevalent SE gene (34%) in all three sources of isolation followed by *seg* (30%) and *sea* (21%). *Ser* was detected in one isolate while *sec, see* and *sej* were not found in all tested strains. Several enterotoxin gene combinations were observed including isolates with a combination of two ($n = 4$, 9%), three ($n = 5$, 11%), four ($n = 13$, 28%) and five ($n = 2$, 4%) different SE genes. There was coexistence of *seo/seg* and *sei/seg* genes (Table 2, Figs. 3 and 4). Eighty-eight percent of t355 ($n = 8$) isolates obtained from two different locations had lukS-PV/lukF-PV and complete absence of all tested SE. Some SE genes were associated

**Table 1  History of bacterial strains and spa types.** BURP spa cluster were named according to possible founder spa types.

| Strain | Spa type | BURP cluster | Town of isolation | State of isolation | Source | Predominant spa type |
|---|---|---|---|---|---|---|
| *S. aureus* FA001 | t355 | Singleton | Ife | Osun | Clinical | t355 (8) |
| *S. aureus* FA002 | t537 | Singleton | Ife | Osun | Clinical | t091 (3) |
| *S. aureus* FA003 | t355 | Singleton | Ife | Osun | Clinical | |
| *S. aureus* FA004 | t355 | Singleton | Ado Ekiti | Ekiti | Clinical | |
| *S. aureus* FA005 | t355 | Singleton | Ado Ekiti | Ekiti | Clinical | |
| *S. aureus* FA006 | t1931 | CC 127 | Ado Ekiti | Ekiti | Clinical | |
| *S. aureus* FA007 | t355 | Singleton | Ado Ekiti | Ekiti | Clinical | |
| *S. aureus* FA008 | t355 | Singleton | Ado Ekiti | Ekiti | Clinical | |
| *S. aureus* FA009 | t355 | Singleton | Ado Ekiti | Ekiti | Clinical | |
| *S. aureus* FA010 | t355 | Singleton | Ado Ekiti | Ekiti | Clinical | |
| *S. aureus* FA012 | t1045 | Singleton | Lagos | Lagos | Clinical | |
| *S. aureus* FA013 | t021 | CC 021 | Lagos | Lagos | Clinical | |
| *S. aureus* FA014 | t069 | CC050 | Lagos | Lagos | Clinical | |
| *S. aureus* FA015 | t1095 | Singleton | Lagos | Lagos | Clinical | |
| *S. aureus* FA016 | t1095 | Singleton | Lagos | Lagos | Clinical | |
| *S. aureus* FA031 | t091 | Singleton | Ife | Osun | Clinical | |
| *S. aureus* FA048 | t091 | Singleton | Ibadan | Oyo | Clinical | |
| *S. aureus* FA049 | t127 | CC 127 | Ibadan | Oyo | Clinical | |
| *S. aureus* FA051 | t127 | CC 127 | Portharcourt | Rivers | Clinical | |
| *S. aureus* FA052 | t008 | CC 1171 | Ibadan | Oyo | Clinical | |
| *S. aureus* FA053 | t091 | Singleton | Ibadan | Oyo | Clinical | |
| *S. aureus* FA034 | t084 | Singleton | Amassoma | Bayelsa | Nasal | t091 (3) |
| *S. aureus* FA035 | t091 | Singleton | Amassoma | Bayelsa | Nasal | t127 (3) |
| *S. aureus* FA036 | t1045 | Singleton | Amassoma | Bayelsa | Nasal | |
| *S. aureus* FA037 | t1045 | Singleton | Amassoma | Bayelsa | Nasal | |
| *S. aureus* FA039 | t127 | CC 127 | Amassoma | Bayelsa | Nasal | |
| *S. aureus* FA040 | t939 | Singleton | Amassoma | Bayelsa | Nasal | |
| *S. aureus* FA041 | t311 | CC 1154 | Amassoma | Bayelsa | Nasal | |
| *S. aureus* FA043 | t127 | CC 127 | Amassoma | Bayelsa | Nasal | |
| *S. aureus* FA044 | t786 | Singleton | Amassoma | Bayelsa | Nasal | |
| *S. aureus* FA045 | t091 | Singleton | Amassoma | Bayelsa | Nasal | |
| *S. aureus* FA046 | t091 | Singleton | Amassoma | Bayelsa | Nasal | |
| *S. aureus* FA047 | t127 | CC 127 | Amassoma | Bayelsa | Nasal | |
| *S. aureus* FA050 | t1154 | CC 1154 | Amassoma | Bayelsa | Nasal | |
| *S. aureus* FA017 | t069 | CC 50 | Ilishan | Ogun | Poultry | t069 (4) |
| *S. aureus* FA018 | t069 | CC 50 | Ilishan | Ogun | Poultry | t091 (2) |
| *S. aureus* FA019 | New | Singleton | Ilishan | Ogun | Poultry | |
| *S. aureus* FA020 | t095 | Singleton | Ilishan | Ogun | Poultry | |

**Table 1** (*continued*)

| Strain | Spa type | BURP cluster | Town of isolation | State of isolation | Source | Predominant spa type |
|---|---|---|---|---|---|---|
| *S. aureus* FA021 | t091 | Singleton | Ilishan | Ogun | Poultry | |
| *S. aureus* FA022 | t069 | CC 050 | Ilishan | Ogun | Poultry | |
| *S. aureus* FA023 | t091 | Singleton | Ilishan | Ogun | Poultry | |
| *S. aureus* FA024 | t292 | CC 1171 | Ilishan | Ogun | Poultry | |
| *S. aureus* FA025 | t939 | Singleton | Ilishan | Ogun | Poultry | |
| *S. aureus* FA026 | t318 | CC 021 | Ilishan | Ogun | Poultry | |
| *S. aureus* FA027 | t069 | CC 050 | Ilishan | Ogun | Poultry | |
| *S. aureus* FA028 | t050 | CC 050 | Ilishan | Ogun | Poultry | |
| *S. aureus* FA029 | t1171 | CC 1171 | Ilishan | Ogun | Poultry | |

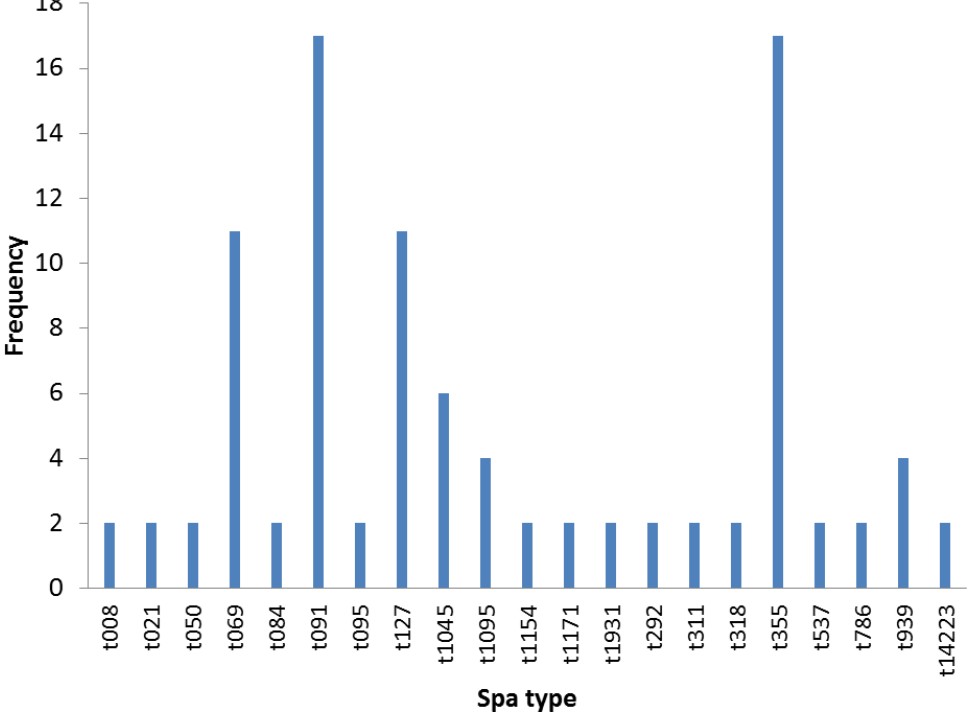

**Figure 1** **Frequency of spa types in 47 *S. aureus* isolates.**

with particular spa types, *sea, seq, seb, sek* were associated with spa type 069 (obtained from 2 locations). All t127 isolates carried *sea.* The only SE most prevalent in spa type t091 was *sep*. MRSA FAA014 had four SE genes while MRSA FAA044, a nasal strain, carried no SE genes.

## DISCUSSION

*We report* predominance of spa type t091 *and relatively high occurrence of SE genes in S. aureus isolates that had been previously collected in Southern Nigeria. Species confirmation of the 47 S. aureus isolates was achieved by femA gene amplification in accordance with* *Vannuffel et al. (1995)* who confirmed that *femA* expression is a unique feature of *S. aureus*,

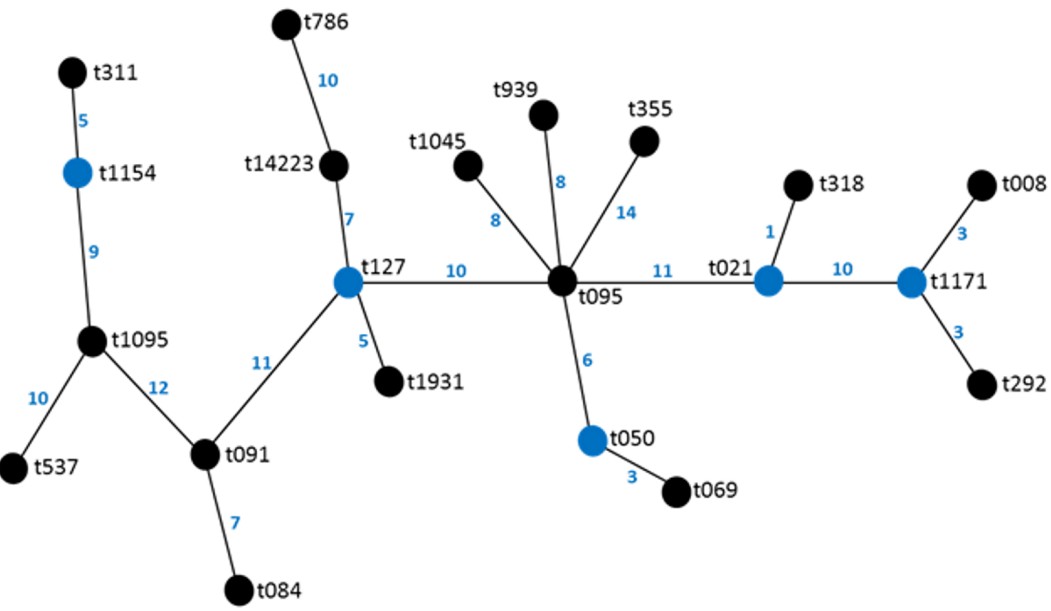

**Figure 2  BURP representation of the spa types.**

allowing its specific detection. We report a relatively low detection of *mecA/mec* in the current study, although several authors had reported high phenotypic detection of MRSA in Nigeria (*Onanuga, Oyi & Onaolapo, 2005*; *Ayeni, Olatunji & Ogunniran, 2014*). *O'Malley et al. (2015)* reported that MRSA exists in clinical and community settings in Nigeria. The two detected MRSA isolates were from clinical and nasal sources with absence of *lukS-PV/lukF-PV* genes. MRSA strains has been previously isolated from nasal swabs of workers in farms (*Macori et al., 2017*). The poultry strains in this study were MRSA negative; however, *Nworie et al. (2017)* has reported MRSA in poultry in Nigeria. Lack of *lukS-PV/lukF-PV* in Nigerian MRSA strains has been previously reported by *Kolawole et al. (2013)*. We, however, observed a relatively high rate of *lukS-PV/lukF-PV* positive isolates in MSSA strains from clinical isolates which are in line with other studies from Africa. Sub-Saharan Africa is observed to be a PVL endemic region showing high PVL prevalence among MSSA isolates. *O'Malley et al. (2015)* indicated that 40% (23/57) of MSSA isolates were *lukS-PV/lukF-PV* positive with no *lukS-PV/lukF-PV* positive MRSA and they concluded that *lukS-PV/lukF-PV* positive isolates are most often seen in MSSA. *Shittu et al. (2011)* also reported high proportion of *lukS-PV/lukF-PV* positive isolates among MSSA (40%) in Nigeria. However, in other region of the world, it has been reported that MSSA rarely harbor *lukS-PV/lukF-PV* (*Becker et al., 2017*).

Molecular typing technologies such as spa typing provide information which enables the grouping of individual isolates in clonal lineages (*Kolawole et al., 2013*). Twenty spa types were found in this study, with five clusters containing at least two spa types and nine singletons. It can be inferred that the spa types were widely different, which is an indication of varied sources of isolation and different geographical locations. However, the highest percentage belonged to t091. This is different from other studies from Nigeria where t064

**Table 2  Characterization and distribution of staphylococci enterotoxins genes in each tested strain.**

| Isolate | Spa type | PVL and Staphylococci Enterotoxins (SE) Genes | | | | | | | | | | | | | | | Total SE no |
|---|---|---|---|---|---|---|---|---|---|---|---|---|---|---|---|---|---|
| | | Pvl | A | O | M | Q | N | K | P | L | B | G | R | U | I | H | |
| *S. aureus* FA001 | t355 | + | | | | | | | | | | | | | | | 1 |
| *S. aureus* FA002 | t537 | | | + | + | | | | | + | | + | | | | | 4 |
| *S. aureus* FA003 | t355 | | | | | | | | | | | | | | | | 0 |
| *S. aureus* FA004 | t355 | + | | | | | | | | | | | | | | | 1 |
| *S. aureus* FA005 | t355 | + | | | | | | | | | | | | | | | 1 |
| *S. aureus* FA006 | t1931 | + | | | | | | | | | | | | | | | 1 |
| *S. aureus* FA007 | t355 | + | | | | | | | | | | | | | | | 1 |
| *S. aureus* FA008 | t355 | + | | | | | | | | | | | | | | | 1 |
| *S. aureus* FA009 | t355 | + | | | | | | | | | | | | | | | 1 |
| *S. aureus* FA010 | t355 | + | | | | | | | | | | | | | | | 1 |
| *S. aureus* FA012 | t1045 | | | + | + | | | | | | | + | | | | | 3 |
| *S. aureus* FA013 | t021 | + | | + | | | | | | | | + | | + | | | 4 |
| *S. aureus* FA014 ** | t069 | | + | | | + | | + | | | + | | | | | | 4 |
| *S. aureus* FA015 | t1095 | | | + | + | | | | | | | + | | | | | 3 |
| *S. aureus* FA016 | t1095 | | | + | + | | | | | | | | | | | | 2 |
| *S. aureus* FA017 * | t069 | | + | | | + | | + | | | + | | | | | | 4 |
| *S. aureus* FA018 * | t069 | | + | | | + | | + | | | + | | | | | | 4 |
| *S. aureus* FA019 * | t14223 | | | | | | | | | | | | | | | | 0 |
| *S. aureus* FA020 * | t095 | | | + | + | | | | | + | | + | | | | | 4 |
| *S. aureus* FA021 * | t091 | | | | | | | | + | | | | | | | | 1 |
| *S. aureus* FA022 * | t069 | | + | | | + | | + | | | + | | | | | | 4 |
| *S. aureus* FA023 * | t091 | | | | | | | | + | | | | | | | | 1 |
| *S. aureus* FA024 * | t292 | | | + | + | | | | | | + | + | + | | | | 5 |
| *S. aureus* FA025 * | t939 | | | + | + | | | | | | | + | | | | | 3 |
| *S. aureus* FA026 * | t318 | + | | + | | | | | | | | + | | + | | | 4 |
| *S. aureus* FA027 * | t069 | | + | | | + | | + | | | + | | | | | | 4 |
| *S. aureus* FA028 * | t050 | | | + | + | | | | | | | + | | | | | 3 |
| *S. aureus* FA029 * | t1171 | | | | | | | | | | | | | | | | 0 |
| *S. aureus* FA031 | t091 | | | | | | | | + | | | | | | | | 1 |
| *S. aureus* FA034 | t084 | | | | | | | | | | | | | | | | 0 |
| *S. aureus* FA035 | t091 | | | + | | | | | + | | | | | | | | 2 |
| *S. aureus* FA036 | t1045 | | | + | | | | | | | | + | | | + | | 3 |
| *S. aureus* FA037 | t1045 | | | + | | | + | | | | | + | | | + | | 4 |
| *S. aureus* FA039 | t127 | + | + | | | + | | + | | | | | | | | + | 5 |
| *S. aureus* FA040 | t939 | | | + | | | + | | | | | + | | | + | | 4 |
| *S. aureus* FA041 | t311 | | | + | | | + | | | | | + | | | + | | 4 |
| *S. aureus* FA043 | t127 | | + | | | | | | | | | | | | | + | 2 |
| *S. aureus* FA044 ** | t786 | | | | | | | | | | | | | | | | 0 |

| Isolate | Spa type | PVL and Staphylococci Enterotoxins (SE) Genes | | | | | | | | | | | | | | | Total SE no |
|---|---|---|---|---|---|---|---|---|---|---|---|---|---|---|---|---|---|
| | | Pvl | A | O | M | Q | N | K | P | L | B | G | R | U | I | H | |
| *S. aureus* FA045 | t091 | | | | | | | | + | | | | | | | | 1 |
| *S. aureus* FA046 | t091 | | | | | | | | + | | | | | | | | 1 |
| *S. aureus* FA047 | t127 | | + | | | | | | | | | | | | + | | 2 |
| *S. aureus* FA048 | t091 | | | | | | | | + | | | | | | | | 1 |
| *S. aureus* FA049 | t127 | | + | | | | | | | | | | | | | | 1 |
| *S. aureus* FA050 | t1154 | | | + | | | + | | | | | + | | | + | | 4 |
| *S. aureus* FA051 | t127 | | + | | | | | | | | | | | | | | 1 |
| *S. aureus* FA052 | t008 | | | | | | | | + | | | | | | | | 1 |
| *S. aureus* FA053 | t091 | | | | | | | | + | | | | | | | | 1 |
| Total No of SE genes | | 11 | 10 | 16 | 8 | 6 | 4 | 6 | 9 | 2 | 6 | 14 | 1 | 2 | 5 | 3 | |
| % | | 23 | 21 | 34 | 17 | 13 | 9 | 13 | 19 | 4 | 13 | 30 | 2 | 4 | 11 | 6 | |

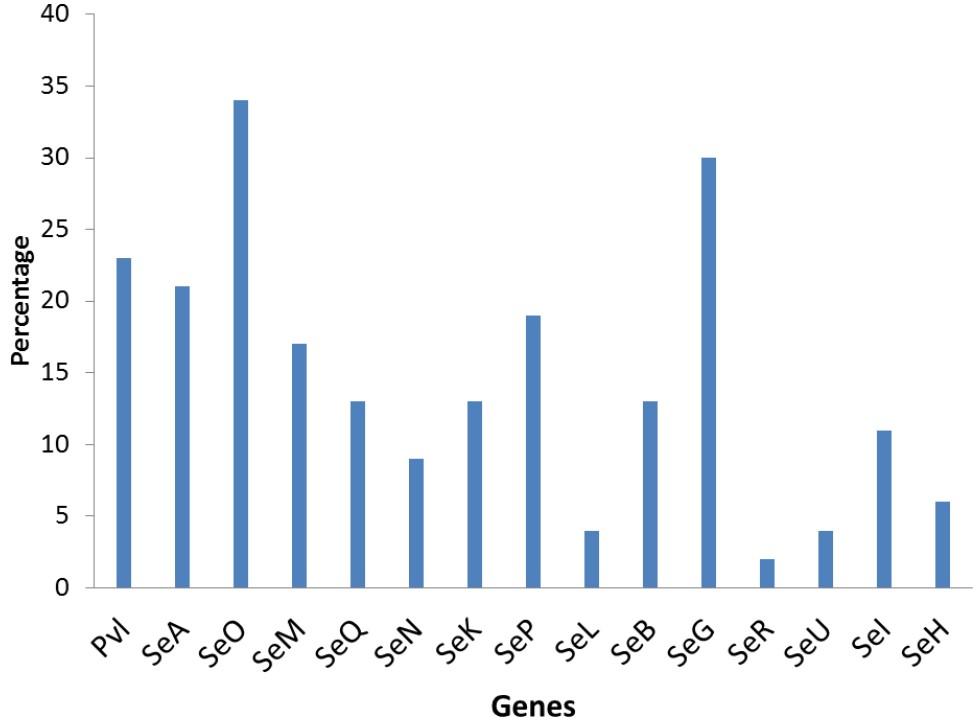

**Figure 3  Prevalence of staphylococci enterotoxin genes in studied isolates.**

had higher prevalence. *Kolawole et al. (2013)* reported the occurrence of 24 spa types with the most frequent spa types being t064, t084, t311 and t1931. Also, spa type t064 is the most common spa type among HIV positive patients in Nigeria (*Olalekan et al., 2012*). *Shittu et al. (2011)* reported a total of 28 spa types with the predominant spa type being t084 among the MSSA isolates, while t451, t008, t002 and t064 were observed in Southwest Nigeria. These studies, however, were confined to Southwestern Nigeria, while a study by
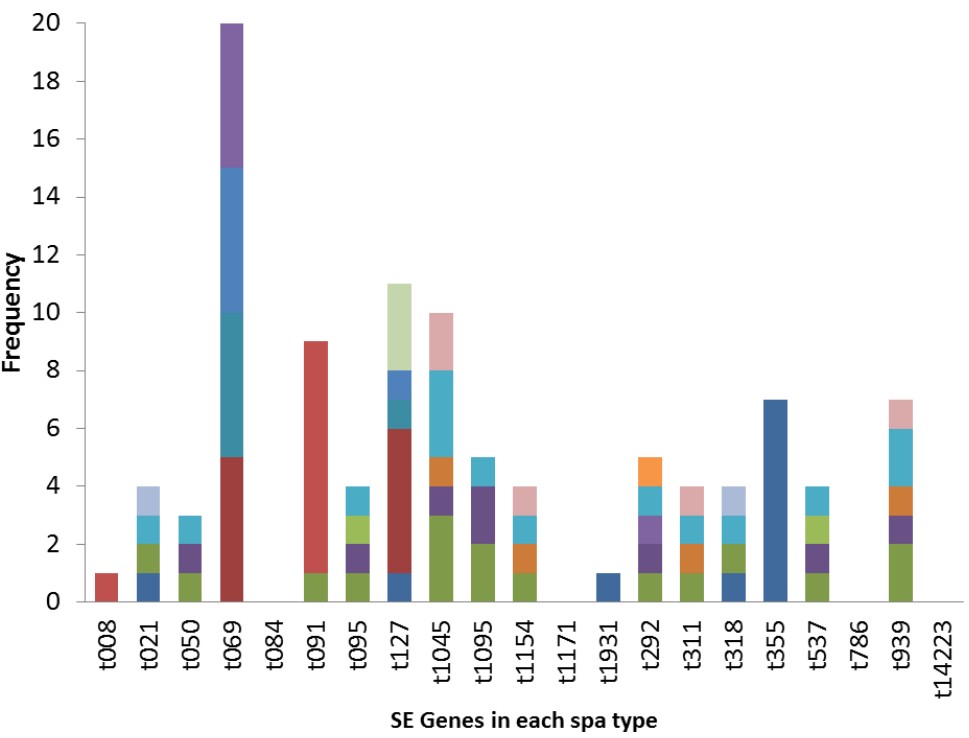

**Figure 4    Association of enterotoxin genes with spa types.**

*O'Malley et al. (2015)* which involved nasal carriage from Southwestern and Southeastern Nigeria reported spa t091 and t355, which we also found in our study. Our study locations were also in the Southwestern and South–South parts of Nigeria and some isolates were from nasal carriages. Therefore, location and site of isolation may be an important factor in *spa* types found. Interestingly, t091 was seen in all three sources of isolation in this study i.e., nasal, clinical and poultry sources. It also spread across widely spaced locations in four states of Southern Nigeria and was consistently seen even in the small number of isolates used in this study. The predominant *spa* type t091 reported in this study has recently been reported in Germany (*Becker et al., 2017*) and Poland (*Ilczyszyn et al., 2016*) while t355 has been recently reported in Uganda (*Asiimwe et al., 2017*) and Italy (*Basanisi et al., 2017*).

It has been observed that prevalence of enterotoxin genes differs greatly depending on the geographic affiliation and the population structure tested (*Kolawole et al., 2013*). In this study, 89% of all tested isolates harbored at least one staphylococcal enterotoxin gene, with *seo* being the most prevalent followed by *seg*. This is a high occurrence and has implications in public health. Staphylococcal enterotoxins may induce T-cell stimulation resulting in systemic illness such as toxic shock syndrome and food poisoning. *Peck et al. (2009)* also reported significant differences and a higher prevalence of selected enterotoxin genes in *S. aureus* isolates obtained from blood compared to nasal isolates (7.2% blood vs. 30.5% nasal). The clinical significance of SE cannot be overemphasized. *Argudín, Mendoza & Rodicio (2010)* stated that staphylococcal food poisoning results from the consumption of foods containing sufficient amounts of preformed enterotoxin and its real incidence

is probably underestimated due to misdiagnosis and improper laboratory examination with the control of social and economic importance. The most characterized SEs are SEA, SEB, SEC, SED and SEE. The genes for these proteins have little or no occurrence in this study. This may be due to the study location which has not been well characterized untill now. Other SEs (SEG, SEH, SEI, SER, SES and SET) have also been identified as potential agents of food poisoning and there are more of these other SE genes in this study. The occurrence of *sen, sei* and *seh* only in nasal isolates has implications in contact contaminations and spread. These strains were previously isolated from students that live together in hostels where there is sharing of many personal items. The genes could therefore be easily spread within this population. Enterotoxin I and H have some roles in food poisoning (*Pinchuk, Beswick & Reyes, 2010*). The main sources of food contamination caused by enterotoxin producing *S. aureus* are food handlers through manual contact via noses and hands (*Denayer et al., 2017*). These could also be applicable to people living closely together in a community as depicted in this study. The high occurrence of SE genes of different types in poultry birds raise concern of the possibility of spread from the infected handlers to the community. *S. aureus* growth and enterotoxin production along the various production chains and final products in poultry should be discouraged because some enterotoxigenic strains with a particular spa type occur in specific products (*Macori et al., 2017*). *Seo* and *sei* were found in association with *seg* in this study. Previous studies have reported associations of *seg* and *sei*. *Kolawole et al. (2013)* reported that the most frequent SE genes detected in different locations were *seg/sei* as reported by in Nigeria by (*Kolawole et al., 2013*), in Norway (*Loncarevic et al., 2005*) and in France (*Rosec & Gigaud, 2002*). The genes *seg* and *sei* were frequently found together because they are within the same cluster, in a 3.2 kb DNA fragment  (*Asiimwe et al., 2017*). The high co-occurrence of *seg* and *sei* found in this study is worrisome because it has been reported that SEG, SEI and SER, possess emetic activities (*Denayer et al., 2017*). *Kim et al. (2011)* reported that *sec, seg, sei, sel, sem, sen, seo*, were associated with genomic islands and could be responsible for their observed combined occurrence. Some *S. aureus* strains in this study also had several enterotoxin gene combinations, from a combination of two to a combination of five different SE genes. Horizontal gene transfer among strains harboring SE genes may not be rare because the genes are located on mobile elements such as prophages, enterotoxin gene clusters, (egc), plasmids, bacteriophages, staphylococcal cassette chromosome (SCC) or pathogenicity islands (*Pinchuk, Beswick & Reyes, 2010*). The presence of an enterotoxin gene may not be a conclusive indication of SE protein expression. However, SE gene screening may be a good tool for the probability of enterotoxins in staphylococcal strains, and the presence of enterotoxin genes in *S. aureus* isolates from healthy carriers highlights the possible risk of food product contamination and spread (*Denayer et al., 2017*). Many isolates with enterotoxin genes different from the classical SEs genes of which the role in food intoxications is not always known were found in this study.

Some SE genes were observed in specific spa types. Most t355 spa types had the *lukS-PV/lukF-PV* gene, in contrast to other *spa* types where there was complete absence of the gene. Spa type t355 is also characterized by complete absence of all investigated SE. *Sea, seq, seb, sek* were observed in *spa* type 069. All t127 carried the *sea* gene, while the *sep* gene was

seen only in spa type t091 and that is the only SE gene that all t091 strains except for one isolate carried. These *S. aureus* strains were isolated from different locations across Nigeria, yet the spa types consistently displayed the presence or absence of a particular SE gene. This information could be useful in predicting virulence toxins that a particular strain of *S. aureus* will likely carries once the spa type is known. However, further representative studies with larger sample sizes are needed to confirm this. *Shittu et al. (2011)* also reported association of some toxin genes (*seh* and *etd*) with a sequence type (ST25).

## CONCLUSIONS

The relatively high prevalence of *seo* and other toxin genes in healthy humans and poultry in this study reveals the potentials of *S. aureus* strains from Nigeria as a potential threat for public health and easy dissemination among strains. This study will also help in tracking the evolution of *S. aureus* epidemic strains in Nigeria and also provide information on some newly described SE genes that lack corresponding phenotypic staphylococcal enterotoxin detection. To the best of our knowledge, this is the first study reporting a high occurrence of staphylococcal enterotoxins genes in poultry from Nigeria.

### Limitation of the Study

This was an explorative study, and limited numbers of samples from nasal, poultry and clinical samples were investigated. Future studies should involve a larger sample size.

## ACKNOWLEDGEMENTS

Bidemi Sunmola is thanked for data analysis.

### Funding

Funmilola A. Ayeni received a 2013/2014 Ernst Mach Postdoctoral Scholarship at the Institute of Medical Microbiology and Hygiene, AGES—Austrian Agency for Health and Food Safety, Spargelfeldstraße, Vienna, Austria. The funders had no role in study design, data collection and analysis, decision to publish, or preparation of the manuscript.

### Grant Disclosures

The following grant information was disclosed by the authors:
Institute of Medical Microbiology and Hygiene, AGES—Austrian Agency for Health and Food Safety, Spargelfeldstraße, Vienna, Austria.

### Competing Interests

The authors declare there are no competing interests.

### Author Contributions

- Funmilola A. Ayeni conceived and designed the experiments, performed the experiments, analyzed the data, prepared figures and/or tables, authored or reviewed drafts of the paper, approved the final draft.

- Werner Ruppitsch conceived and designed the experiments, analyzed the data, prepared figures and/or tables, authored or reviewed drafts of the paper, approved the final draft.
- Franz Allerberger conceived and designed the experiments, contributed reagents/-materials/analysis tools, authored or reviewed drafts of the paper, approved the final draft.

## Data Availability

The raw data are provided in Data S1.

## Supplemental Information

Supplemental information for this article can be found online at http://dx.doi.org/10.7717/peerj.5204#supplemental-information.

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
