# Peer review of "Molecular characterization of clonal lineage and staphylococcal toxin genes from S. aureus in Southern Nigeria"

_PeerJ, doi:10.7717/peerj.5204_

## Round 0.1 · original submission · Major Revisions

This manuscript is borderline between reject and major revision due to the large number of aspects that need addressing. However, I would like to give you the opportunity to address all of the comments of the reviewers and therefore will allow for a resubmission.

I would draw your attention in particular to reviewer 2's comments who has particular expertise in this field. It is important for you to highlight the exact aim of the study and use the manuscript to then show how your methodology will answer your research question. Many of the techniques used are very standard and more experiments are required, as detailed by reviewer 2. The lack of controls is also an important issue, with a particular need to place the enterotoxin gene typing in context. More background is also needed to determine why you have chosen particular elements to focus on in the study.

Before re-submission please make sure you thoroughly proof-read the manuscript and ensure that simple things like differentiating between proteins and gene names is performed.

Reviewer 1 ·

Basic reporting

Tables are not well presented. Especially table 2. No title provided and some contents are written in short forms and no legends to define them. In the text, it has been indicated that 2 MRSA isolates were detected, and in the table only one is indicated.
I would suggest to have one more table that provides social demographic information of the the clinical S.aureus isolates, perhaps poultry too. from the table it should be indicated dates of isolation, source and health status (in case of human s.aureus) and if the patient was admitted or not. this is important as it might tell the which spa types are related to certain conditions,,but also we can see if certain spa types are related with hospital environment or community

Experimental design

Since it has been indicated that t064 was more prevalent in HIV individuals from previous studies ; it was wealth in the current study to include the suggested . this information will provide information on spa types in relation to disease condition. I understand it is not the main aim,but this will add value to this article .

Validity of the findings

The conclusion is not well stated.This conclusion repeats the results. It would be better to give out what you think this report has contributed and importance of this report to the locality of the study and globally

Additional comments

The manuscript is about: Molecular characterization of clonal lineage and staphylococcal toxin genes from S. aureus in Southern Nigeria.
It is well written.
The data is very important but you must state this locally generated data will offer to the settings. considering that this kind of methodology is not part of routine diagnostics in resource limited settings. by having this information what do you think are the benefits?

Reviewer 2 ·

Basic reporting

The authors typed 47 Staphylococcus aureus strains for staphylococcal enterotoxin genes previously presented in other studies in Southern Nigeria. The aim of this study was to characterize the S. aureus isolated from human and poultry by determining spa types and the presence/absence of enterotoxin genes, mecA/C, and pvl.
I appreciate the authors focused their study on such a subject, but there are many questions and problems remaining to be clarified.

1. Abstract. An abstract must be a concise synopsis of the key parts of the research, at least contains a brief conclusion summary. There are no any study conclusions that satisfy the criteria of the journal;
2. On bacterial strains. Isolation history and detailed profiles of strains whose “lineages” was determined are important for better understanding or interpreting the information presented;
What is the relation between the poultry strains and the humans? Why have been chosen these strains? The authors should provide detailed information about the isolates used in this study, including details of the carriers, isolation history etc. (year, date, city, country, etiological foods if it was a food-borne outbreak, fatalities, etc);
3. Nowadays, there are many techniques for typing S. aureus. The methods presented are not sufficient for the “clonal lineage” study. In fact the author stressed about this term but there is no genetic study clearly presented. Would be interesting present a minimum spanning tree with hypotetical node computing the spa-types. A comparative analysis with the strains presented is necessary to support some conclusions.
Other phenotypic or genotypic typing methods must be used to discriminate the different S. aureus isolates, for host attribution or to determine the relatedness of different isolates.
In addition, the analysis of the polymorphic X region of the protein A gene (spa typing) was presented in detail. It is sufficient to cite the paper from Koreen et al., (2004) for reference. The same for the other methods cited, as soon as they are not novel or presented and used with variation on the protocol.
4. The authors carried out staphylococcal enterotoxin genes typing, this is a very important point of the study but many information of enterotoxins was lacked or wrong. This point is critical and cannot be overlooked. The authors’ data lacks reliability due to the fact that there were no positive and negative controls mentioned in the manuscript for all the analysis performed. Another problem is lack of information about the previous study cited as the master paper as source of the isolates.
5. Genetic analysis was not enough. It is not clear the reason why the virulence factors included were considered important for the conclusion of the study. In addition the term used are not correct and there are many errors in the classification of the toxins.
The authors focused their study on the toxin genes, as seen in their title. But many information of enterotoxins was lacked or wrong. This point is critical and cannot be overlooked.

Conclusion.
The conclusions are not clear and there is not relevance for the few data presented. All genes of enterotoxin except for SElX, SElY and SElZ are located on mobile genetic elements or genetic clusters, and are highly variable, which might be closely relevant with their dynamic evolution (Sato’o et al. 2014 JCM 52:2637-2649, Suzuki et al. 2014. MAI. 58:570-580, Suzuki et al. 2015. JAM 118:1507-1520. Suzuki et al. 2017. Int J Food Microbiol. 262:31-37). The authors must give more space in the discussion for this subject.

Evaluation scheme:
1. The language is not clear and several time is ambiguous, intense professional English editing is needed.
2. The article must be written in English and must use clear, technically correct text. The article lacks in this standard. Strong conforming process is required.
3. The Literature references are not sufficient. The authors cannot avoid citing the most recent relevant papers published on MRSA and enterotoxins for support the area of research they speculated and give a context to support their conclusions.
The article should include detailed introduction and background to demonstrate how the work fits into the broader field of staphylococcal enterotoxins. Relevant prior literature should be appropriately referenced.
4. The manuscript is not well structured and there are several formatting errors, including on figures and tables. The raw data shared as a supplemental file is not clear on the information presented. What toxins the authors referred to? Are maybe referring to the genes instead?
5. The structure of the article is conform to an acceptable format of ‘standard sections’, but it clearly need a formatting editing.

Experimental design

The experiment is very easy but the methods are described in a useless way.
The investigation must have been conducted rigorously, in this case very low standards have been followed that do not meet the technical standard. The analysis must include negative and positive controls at all levels.

Validity of the findings

The data are very poor and there is not a clear representation of the results. The conclusions are not well stated and there are limited supports results.

---

## Round 0.2 · Minor Revisions

Please note that the English language and grammar used throughout the manuscript is still not at an acceptable enough standard for publication. Please can the manuscript be thoroughly reviewed by a native speaker prior to re-submission. In addition, please see the additional notes by reviewer 2.

Reviewer 2 ·

Basic reporting

Language not clear and ambiguous use of the English.
Literature reference must be improved, in particular in the discussion section (see notes for the authors).

Experimental design

No comment

Validity of the findings

Minor revision and more information related to the speculation of the impact and the novelty of the study.

Additional comments

Suggest a discussion that take in consideration the already published studies:
About the spread of the MRSA in poultry:
- Bhedi KR, Nayak JB, Brahmbhatt MN, Roy A, Mathakiya RA, Rajpura RM. Detection and Molecular Characterization of Methicillin-Resistant Staphylococcus aureus Obtained from Poultry and Poultry House Environment of Anand District, Gujarat, India. International Journal of Current Microbiology and Applied Sciences Vol. 7 No. 2018;2:867-72.

About MRSA in ovine:
- Macori G, Giacinti G, Bellio A, Gallina S, Bianchi DM, Sagrafoli D, Marri N, Giangolini G, Amatiste S, Decastelli L. Molecular epidemiology of methicillin-resistant and methicillin-susceptible Staphylococcus aureus in the ovine dairy chain and in farm-related humans. Toxins. 2017 May 16;9(5):161.

About the approach for a population structure study, correlation between spa-types and a food-chain sector.
- Johler S, Macori G, Bellio A, Acutis PL, Gallina S, Decastelli L. Characterization of Staphylococcus aureus isolated along the raw milk cheese production process in artisan dairies in Italy. Journal of dairy science. 2018 Apr 1;101(4):2915-20.

---

## Round 0.3 · Minor Revisions

I read the first few sentences of this resubmission and found very similar mistakes that have been pointed out in the past. S. aureus is never expanded upon, the grammar is poor and it needs a complete re-read and check. Please do not re-submit unless the manuscript has been fully copy-edited.

---

## Round 0.4 · accepted · Accept

Thank you for addressing the minor errors and improving the readability of this manuscript.

#